# Combined TLR-3/TLR-8 Signaling in the Presence of α-Type-1 Cytokines Represents a Novel and Potent Dendritic Cell Type-1, Anti-Cancer Maturation Protocol

**DOI:** 10.3390/cells11050835

**Published:** 2022-02-28

**Authors:** Tadej Fevžer, Primož Poženel, Kaja Zajc, Nataša Tešić, Urban Švajger

**Affiliations:** 1Division of Gynaecology and Obstetrics, University Clinical Center Ljubljana, Šlajmerjeva 3, SI-1000 Ljubljana, Slovenia; fevzer.tadej@gmail.com; 2Blood Transfusion Center of Slovenia, Šlajmerjeva 6, SI-1000 Ljubljana, Slovenia; primoz.pozenel@ztm.si (P.P.); kaja.zajc@ztm.si (K.Z.); natasa.tesic@ztm.si (N.T.); 3Faculty of Pharmacy, University of Ljubljana, Aškerčeva 7, SI-1000 Ljubljana, Slovenia

**Keywords:** dendritic cells, maturation, type-1 polarization, cytotoxic T cells

## Abstract

During the ex vivo generation of anti-cancer dendritic cell (DC)-based vaccines, their maturation still represents one of the most crucial steps of the manufacturing process. A superior DC vaccine should: possess extensive expression of co-stimulatory molecules, have an exceptional type-1 polarization capacity characterized by their ability to produce IL-12p70 upon contact with responding T cells, migrate efficiently toward chemokine receptor 7 (CCR7) ligands, and have a superior capacity to activate cytotoxic T cell responses. A major advance has been achieved with the discovery of the next generation maturation protocol involving TLR-3 agonist (poly I:C), tumor necrosis factor (TNF)-α, interleukin (IL)-1β, interferon (IFN)-γ, and IFN-α, and has since been known as α-type-1 maturation cocktail. We demonstrate how this combination can be greatly enhanced by the inclusion of a TLR-8 stimulation (R848), thereby contributing to potentiation between different TLR signaling pathways. For maximum efficiency, TLR-3 stimulation should precede (termed pre I:C) the stimulation with the R848/TNF-α/IL-1β/IFN-α/IFN-γ cocktail. When compared to DCs matured with α-type-1 maturation cocktail (αDCs), DCs matured with pre I:C/R848/TNF-α/IL-1β/IFN-α/IFN-γ (termed zDCs) displayed higher expression of CD80 and CD86 co-stimulatory molecules. Importantly, after CD40-ligand stimulation, which simulates DC-T cell contact, zDCs were much more proficient in IL-12p70 production. In comparison to αDCs, zDCs also displayed a significantly greater migratory capacity toward chemokine ligands (CCL)19 and CCL21, and had a significantly greater allo-stimulatory capacity. Finally, zDCs were also superior in their capacity to induce melanoma-specific CD8+ T cells, CD8+ T cell proliferation, and cytotoxic T cells, which produced approximately two times more IFN-γ and more granzyme B, than those stimulated with αDCs. In conclusion, we present a novel and superior DC maturation cocktail that could be easily implemented into next generation DC vaccine manufacturing protocols in future trials.

## 1. Introduction

Dendritic cells (DCs) are regarded as one of the primary biological tools for use in cancer immunotherapy. Their capacity to induce powerful antigen-specific cytotoxic T cell responses is unprecedented in the immune system [1]. Although their past clinical success as cellular vaccines has been modest at best [2], the variety of mechanisms that underlie their ability to efficiently uptake and present tumor-associated antigens (TAAs) still leaves a lot of opportunity for improvement. 

Out of several different subsets that make up the DC compartment, namely conventional DCs (type-1 and type-2), plasmacytoid DCs, and monocyte-derived DCs (MoDCs), ex vivo generated MoDCs are still most widely utilized due to several operational advantages, such as the ability to obtain large numbers of cells that can be easily manipulated, e.g., cryopreserved for future use in case of repetitive administrations [3]. The manufacture of MoDC-based anti-tumor vaccines generally consists out of four basic steps: patient apheresis; monocyte selection; monocyte-to-DC differentiation using granulocyte-macrophage colony stimulating factor (GM-CSF) and interleukin (IL)-4; and DC maturation along with TAA loading [4]. While the first three steps are more or less similar across various protocols, the maturation of DCs has varied greatly in the last few decades and most likely plays a major role in the clinical success of DC vaccines [5].

The development in this context ranged from early stages, where DCs were used in their immature state (first generation vaccines), then later activated (second generation vaccines) with tumor necrosis factor (TNF)-α, or a combination of multiple inflammatory cytokines. During this time, a «gold standard» cocktail was established that consisted of TNF-α, IL-1β, IL-6, and prostaglandin E2, and was used in numerous in vitro and in vivo studies [6]. The next generation protocols that have been developed in recent years (from 2004 onward) greatly enhance several major DC characteristics important in anti-tumor immunity [7,8].

Among next generation DC maturation cocktails, the work of Kalinski et al. presented one of the most important contributions, demonstrating that combination of toll-like receptor (TLR)-3 stimulation with type I and type II interferons (IFNs) greatly enhances the key anti-tumor DC characteristics, namely antigen presentation, migration in response to chemokine receptor (CCR)-7 ligands, and their capacity to produce IL-12p70 upon CD40-ligand (CD40L) stimulation. This has been designated as α-type-1 polarization cocktail, and consists of polyinosinic: polycytidylic acid (poly I:C), TNF-α, IFN-α, IFN-γ, and IL-1β [7]. This was the first time that type I IFNs were introduced into a DC maturation protocol, which, in combination with IFN-γ (a type II IFN), resulted in a greatly enhanced production of IL-12p70 by DCs [9].

The increased production of IL-12p70 is a hallmark of a quality anti-tumor DC vaccine, since IL-12p70 orchestrates the polarization of CD4+ T cells toward Th1 helper cells during activation, and supports the development and activation of cytotoxic CD8+ T cell (CTL) effectors that kill target tumor cells in an Ag-specific manner [10,11]. The capacity of MoDCs to produce IL-12p70 is not only greatly dependent on the quantitative extent of their maturation, but also on the specific nature of their maturation, as defined by distinct maturation signals. This is clearly demonstrated by the unique increase in IL-12p70-producing capacity following TLR and cytokine signaling in the above-mentioned α-type-1 polarization cocktail. Furthermore, the importance of IL-12p70 production is time-dependent, being most critical directly during DC-T cell contact to affect T cell polarization.

Improved DC maturation characteristics can also be achieved between different TLR pathways [12]. More recently, it has been proposed that prior stimulation via TLR3 (e.g., using poly I:C) can subsequently augment the signaling of other nucleic acid sensing (NAS) TLRs (TLR7, TLR8 and TLR9) by upregulating a specific accessory protein UNC93B1 [13]. Since, allegedly, UNC93B1 also causes the stabilization and increased activation of TLR8, MoDCs, which are known to express TLR8 (in contrast to TLR7 and TLR9, which are expressed by plasmacytoid DCs), could represent the target for appropriate manipulation by TLR agonists. Based on these findings, we set out to investigate whether the additional optimization of the type-1 DC maturation program can be achieved by exploiting the TLR signaling reinforcement between prior TLR3 and subsequent TLR8 stimulation in combination with α type-1 cytokines.

## 2. Materials and Methods

### 2.1. Cell Culture and Isolation

Buffy coats from venous blood of normal healthy volunteers were obtained from the Blood Transfusion Centre of Slovenia, according to institutional guidelines and with the approval of the National Medical Ethics Committee under approval number 0120-279/2017-3. Peripheral blood mononuclear cells (PBMCs) were isolated using Lympholyte^®^-H (Cedarlane laboratories, Burlington, NJ, USA). The cells were washed twice with Dulbecco’s phosphate-buffered saline (DPBS), counted, and used as a source for immunomagnetic isolation of CD14-positive cells (Miltenyi Biotec GmbH, Bergisch Gladbach, Germany). The purity of monocytes was always greater than 90%, as determined by flow cytometry. For monocyte-to-DC differentiation, we used Cellgenix^®^ DC GMP medium (Cellgenix GmbH, Freiburg, Germany), serum-free, supplemented only with 50 µg/mL gentamicin. In addition, 800 U/mL of rhGM-CSF and 1000 U/mL of rhIL-4 (both Peprotech, London, UK) were added to cell culture. On day 2 and 4, half of the medium was exchanged along with starting quantities of rhGM-CSF and IL-4. On day 6, the differentiated monocytes (immature DCs) were harvested, washed twice with DPBS, and counted using Vi-CellTM XR cell viability analyzer (Beckman Coulter, Fullerton, CA, USA). For further analyses or experiments, the DCs were either used immature or matured with the following maturation cocktails, when applicable: LPS (20 ng/mL)/IFN-γ (1000 U/mL); poly I:C (20 µg/mL); R848 (2.5 µg/mL); poly I:C/R848; pre I:C/R848 (poly I:C added 2 h before R848); poly I:C/R848/TNF-α (1000 U/mL)/IL-1β (25 ng/mL)/IFN-α (3000 U/mL)/IFN-γ; or pre I:C/R848/TNF-α/IL-1β/IFN-α/IFN-γ. Poly I:C and R848(resiquimod) are well known TLR3 and TLR8 agonists, respectively, and were purchased from Invivogen^®^ (Toulouse, France). As documented by the manufacturer (www.invivogen.com, accessed on 13 December 2021), the TLR3 agonistic acitivity (EC_50_) for poly I:C is 82 ± 8 ng/mL, while R848′s agonistic acitivity is in the 10 ng–10 µg/mL range (as determined by TLR8 reporter assay). Interleukin-1β, IFN-α2b, and IFN-γ were all purchased from Peprotech (London, UK). For functional assays and phenotypic analysis, the DCs were matured for 24 h in all cases. The timeline of various DC’s maturation schemes is depicted in Figure 1C.

T cells were purified from human buffy coats. Whole CD4+ T cells were obtained by positive selection using CD4 microbeads (Miltenyi Biotec, Bergisch Gladbach, Germany). The purity of CD4+ cells was always >90%, as determined by flow cytometry. Naïve CD4+CD45RA+ T cells were isolated using the naïve CD4+ T cell isolation kit (Miltenyi Biotec), strictly following the manufacturer’s protocol. The purity of isolated naïve CD4+ T cells was always >95%. Whole CD8+ T cells were isolated using CD8 microbeads (Miltenyi Biotec, Bergisch Gladbach, Germany). The purity of isolated CD8+ T cells was always greater than 90%.

### 2.2. Phenotypic Characterization

For flow cytometry analysis of surface DC phenotype, we used the following monoclonal antibodies (mAb): FITC-labeled anti-CD1c (Biolegend, San Diego, CA, USA), anti-CD14, anti-CD16 and anti-HLA I (all from Invitrogen, Camarillo, CA, USA), Alexa Fluor 488-labeled anti-CD1a and anti-CCR7 (both Biolegend). PE-labeled antibodies included: anti-CD11b, anti-CD11c (both Biolegend), anti-CD80, anti-CD86, anti-DC-SIGN, (all Biolegend), and anti-HLA DR (Miltenyi Biotec, Bergisch Gladbach, Germany).

DCs differentiated in Cellgenix^®^ DC GMP medium (Cellgenix GmbH, Freiburg, Germany) were harvested and collected by centrifugation. Before staining, the cells were washed twice in DPBS in all cases. Antibody was added and the cells were incubated for 15 min in the dark, then washed twice and resuspended in 2% paraformaldehyde (PHA). Samples were analyzed on a FACSCalibur system (BD biosciences, Franklin Lakes, NJ, USA). Data were analyzed with CellQuest software (BD biosciences).

### 2.3. Detection of IL-12p70 Production

Dendritic cells were differentiated and subsequently matured using various protocols, as described above. Their capacity to produce IL-12p70 was measured in two ways. In the first, culture supernatants were taken directly from the cultures after 24 h maturation. In the second, the DCs were thoroughly washed after 24 h of maturation, and then re-stimulated via CD40-CD40L pathway using the CD40-ligand Multimer Kit (Miltenyi Biotec, Bergisch Gladbach, Germany) for an additional 24 h, to simulate DC-T cell interaction, as demonstrated previously [14]. IL-12p70 in culture supernatants was determined by a sandwich enzyme-linked immunosorbent assay (ELISA) method using Biolegend’s ELISA MAXTM Deluxe Set for IL-12p70. All assays were set up in duplicate and performed according to manufacturer’s instructions. The lowest detection limit for IL-12p70 was 4 pg/mL.

### 2.4. Detection of Secreted IFN-γ

We determined IFN-γ levels secreted from CD4+ Th1 effectors or CD8+ T cells. For determination of Th1 responses, we stimulated naïve CD4+CD45RA+ T cells using either immature DCs, αDCs or zDCs. Co-cultures were performed in 48-well tissue plates, each well containing 1 × 10^6^ naïve CD4+ T cells, 1 × 10^5^ corresponding DCs as stimulators, and RPMI 1640, supplemented with 10% human AB serum as culture medium. After 7 days, the T cells were collected, washed, re-plated, and re-stimulated via their T cell receptor using T cell activation/expansion macrobeads (anti-CD2/CD3/CD28) from Miltenyi Biotec. After 24 h, the T cell supernatants were taken and levels of IFN-γ were measured using the cytokine bead array method (CBA assay, BD Biosciences) according to manufacturer’s protocol. For CD8+ T cells, co-cultures were performed in 48-well tissue plates, each well containing 1 × 10^6^ positively selected CD8+ T cells and 1 × 10^5^ variously stimulated DCs. After 5 days, supernatants were taken directly from the co-cultures and analyzed for the presence of IFN-γ using the CBA assay (BD Biosciences). The results were analyzed on FacsCalibur flow cytometer.

### 2.5. DC Migration Assay

For assessment of DC migration capacity, we used the Transwell^®^ system (Corning, NY, USA) with 8.0 µm polycarbonate filter inserts (8.0 µm pore size). Optimally differentiated and variously activated DCs were collected and washed twice with DPBS and then re-suspended in Cellgenix^®^ DC GMP medium. The cells were then seeded in the upper compartments of the transwell system in 100 µL. The lower compartments contained 500µL medium with the addition of 200 ng/mL of chemokine CCL21 (Peprotech, London, UK). The plate was incubated for 3 h at 37 °C, 5% CO_2_. Afterwards, the transwell inserts were lifted and the migrated cells from the lower compartments carefully collected. The numbers of migrated cells were counted by flow cytometry using 60 s counts on a FACSCalibur flow cytometer (BD biosciences), as described previously [15].

### 2.6. T Cell Proliferation Assay

Dendritic cells differentiated in Cellgenix^®^ DC GMP medium were used as stimulators in co-culture experiments with allogeneic T cells. The DCs were either left in their immature state or were activated using poly I:C/TNF-α/IFN-α/IFN-γ/IL-1β (αDCs) or using the protocol pre I:C/R848/TNF-α/IFN-α/IFN-γ/IL-1β (zDCs). Purified whole CD4+ T cells or purified whole CD8+ T cells were used as responders. The assays were carried out in 96-well plates, with a total volume of 200 μL per well, and 2 × 10^4^ and 2 × 10^5^ cells were used for DCs and responder T cells, respectively. After 4 days, the wells were pulsed with 1 μCi/well 3H-thymidine (Perkin Elmer, Boston, MA, USA) and proliferation measured by its incorporation after 18–20 h by liquid scintillation counting.

### 2.7. Intracellular Flow Cytometry Assays

For determination of intracellular cytokines, we performed co-cultures between variously stimulated DCs and naïve CD4+CD45RA+ T cells. The co-cultures were performed in 48-well tissue plates in RPMI 1640, supplemented with 10% human AB serum (at 1:10, DC:T cell ratio). After 7 days, the T cells from the co-cultures were collected, washed, and re-stimulated using phorbol myristate acetate (PMA) and anti-CD2/CD3/CD28 macrobeads for 2 h. Afterwards, Brefeldin A was added to the co-cultures, and the cells were further stimulated for an additional 4 h. After stimulation, the cells were collected, washed, fixed, and permeabilized using fixation and permeabilization buffers (Biolegend, San Diego, CA, USA). Then, the cells were stained with PE-conjugated anti-IL-2, anti-IL-10 mAbs, and with FITC-conjugated anti-IFN-γ mAb (all from Miltenyi Biotec, Bergisch Gladbach, Germany).

For intracellular granzyme B determination, we performed co-cultures between variously stimulated DCs (iDCs, αDCs, and zDCs) and positively selected, whole CD8+ T cells, as described previously [16]. On day 5 of co-culture, the T cells were collected, washed, and stained using APC-conjugated anti-CD8 mAb. Afterwards, the cells were fixed and permeabilized using fixation and permeabilization buffers (Biolegend, CA, USA). The cells were then stained intracellularly using PE-conjugated anti-granzyme B mAb (Biolegend, CA, USA). The results were analyzed on MACSQuant 10 flow cytometer (Miltenyi Biotec).

### 2.8. Induction of Melanoma Ag-Specific CD8+ T Cell Responses

Whole CD8+ T cells (purity > 90%) autologous to monocytes used for DC differentiation, were isolated from HLA-A2+ buffy coats by positive immunomagnetic selection (CD8 microbeads, Miltenyi Biotech, Bergisch Gladbach, Germany). CD8+ T cells (2 × 10^6^ cells) were sensitized by autologous iDCs, αDCs, or zDCs (2 × 10^5^), pulsed with four melanoma-associated, HLA-A2-restricted Ag peptides (10 µg/mL per peptide, for 3 h): gp100 (154–162, KTWGQYWQV); gp100 (209–217, ITDQVPFSV); tyrosinase (369–377, YMDGTMSQV); and melan-A (26–35, ELAGIGILTV) (all from Panatech, Heilbronn, Germany). The co-culture was performed for 14 days. On day 0 and 7, we added 10 ng/mL of IL-7, and 50 U/mL of IL-2 (both Peprotech, London, UK) on day 3 and day 10. On day 7, the T cells were re-stimulated with DCs in the identical manner as for the first stimulation. On day 14, the T cells were harvested and re-plated (1 × 10^5^ cells) on pre-coated IFN-γ ELISPOT strips (AID Autoimmun Diagnostika GmbH, Strassberg, Germany). The cells were then re-stimulated with T2 cell line (ATCC) as stimulator cells (5 × 10^4^ cells), which were pulsed in advance with the same four melanoma-associated peptides as DCs. The number of spot-forming colonies was evaluated using ELISPOT reader (AID Autoimmun Diagnostika, Strassberg, Germany).

### 2.9. Statistical Analysis

Statistical analysis was performed using the software Graphpad Prism^®^ version 6.07 for Windows (Graphpad, San Diego, CA, USA). Statistical significance between individual pairs was calculated using Student’s unpaired *t* test. A *p* value of <0.05 was considered statistically significant. Assumption of normality of data distribution was confirmed by performing the Kolmogorov–Smirnov test of normality.

## 3. Results

### 3.1. Prior Stimulation of TLR3 Followed by Stimulation of TLR8 in Combination with α-Type-1 Cytokines Results in Extensive Capacity of DCs to Produce IL-12p70 upon CD40 Ligation in Comparison to Controls

We differentiated DCs from peripheral blood monocytes as described in the Materials and Methods section. On day 6, the immature DCs were harvested and characterized as CD1ahigh, CD14low, or CD209high (data not shown). To determine maturation protocol-dependent IL-12p70 producing capacity, we initially performed a broad comparison analysis. Dendritic cells were washed extensively and left untreated (GM-CSF only), or were subjected to the following maturation protocols: poly I:C/TNF-α/IFN-α/IFN-γ/IL-1β (termed αDCs); LPS/IFN-γ; poly I:C; R848; poly I:C/R848; 2 h pre-treatment with poly I:C and subsequent addition of R848 (pre I:C/R848); simultaneous treatment via TLR3/TLR8 and with α-type-1 cytokines, poly I:C/R848/TNF-α/IFN-α/IFN-γ/IL-1β; or 2 h pre-treatment with poly I:C and subsequent stimulation via TLR8 with α-type-1 cytokines, pre I:C/R848/TNF-α/IFN-α/IFN-γ/IL-1β (termed zDCs). The overall duration of DC stimulation was 24 h in all cases. After maturation, culture supernatants were analyzed for the presence of IL-12p70 (Figure 1A). Additionally, DCs from the cultures were washed extensively, re-seeded, and stimulated using CD40-ligand Multimer kit (Milteny Biotec) for 24 h. Afterwards, the supernatants from CD40-ligand stimulated cultures were analyzed for the presence of IL-12p70 (Figure 1B). When measured directly from supernatants of maturing DCs, the levels of IL-12p70 were significantly higher for zDCs (pre I:C/R848/TNF-α/IFN-α/IFN-γ/IL-1β) in comparison to αDCs (poly I:C/TNF-α/IFN-α/IFN-γ/IL-1β), with an approximately two-fold greater production capacity. Without the addition of α-type-1 cytokines, a potentiation effect between TLR3 and TLR8 could be seen in comparison to TLR3 stimulation alone, particularly when poly I:C was used as a pre-treatment. However, the addition of α-type-1 cytokines greatly induced the capacity of DCs to produce IL-12p70. More importantly, the IL-12p70-producing capacity upon CD40-ligand stimulation was greatly enhanced in zDCs compared to αDCs, being more than eight-fold greater (Figure 1B). Notably, the effect of using poly I:C as a pre-treatment resulted in four-fold greater IL-12p70-producing capacity upon CD40-ligand stimulation, compared to simultaneous TLR3 and TLR8 stimulation using the poly I:C/R848/TNF-α/IFN-α/IFN-γ/IL-1β cocktail.

### 3.2. zDCs Display an Extensive Mature Phenotype, Comparable to αDCs

Dendritic cells were differentiated from peripheral blood monocytes, as described in the Materials and Methods section. On day 6, the obtained immature DCs were either left untreated or were stimulated with α-type-1 polarization cocktail (poly I:C/TNF-α/IFN-α/IFN-γ/IL-1β (αDCs)), combined simultaneous TLR3 and TLR8 stimulation (poly I:C/R848), combined TLR3 and TLR8 simulation with TLR3 pre-treatment (pre I:C/R848), TLR3 and TLR8 stimulation with α-type-1 cytokines (poly I:C/R848/TNF-α/IFN-α/IFN-γ/IL-1β), or were pre-treated with poly I:C for 2 h, followed by additional stimulation with R848/TNF-α/IFN-α/IFN-γ/IL-1β (zDCs). The complete period of maturation was 24 h in all cases. Afterwards, we performed a surface phenotype analysis of major DC co-stimulatory molecules and measured the expression of HLA-DR and CCR7. In general, the expression of the CD86 co-stimulatory molecule, as well as that of HLA-DR and CCR7, was comparable between αDCs and zDCs (Figure 2). In the case of CD80, it’s expression was greater on zDCs than on αDCs (Figure 2). In general, with specific regard to the co-stimulatory phenotype, all maturation approaches enabled a similar extent of DC maturation. 

### 3.3. Migratory and Allo-Stimulatory Capacities of zDCs Are Superior to αDCs

To determine the capacity of variously stimulated DCs to migrate in response to CCR7 ligand CCL21, we performed a transwell assay, as described in the Materials and Methods section. For this purpose, we used either unstimulated, immature DCs for controls, which we compared to αDCs, zDCs, or DCs stimulated with a combination of LPS and IFN-γ. After 3 h of migration period, the migrated cells were collected from the lower chambers and counted by a flow cytometer using 60 s counts (Figure 3A). In comparison, zDCs migrated most efficiently toward CCL21, and were significantly more efficient than αDCs. The migration of DCs stimulated with LPS/IFN-γ was comparable with αDCs. For measurement of DC allo-stimulatory capacity, we used differentially stimulated DCs in a mixed lymphocyte reaction co-culture with allogeneic whole CD4+ T cells. For stimulators, we used either immature DCs, αDCs, or zDCs. Co-cultures were performed in 96 flat bottom wells with a DC:T cell ratio of 1:10 (2 × 10^4^ DCs and 2 × 10^5^ T cells, in each well). Cell proliferation was measured on day 5 by liquid scintillation counting (Figure 3B). As expected, αDCs possessed extensive allo-stimulatory capacity in comparison to immature DCs (cpm counts 32,655 ± 14,466 and 87,286 ± 12,033 for iDCs and αDCs, respectively). However, zDCs had an even stronger capacity to induce T cell proliferation (cpm counts 115,879 ± 12,211), which was significantly higher than that of αDCs.

### 3.4. zDCs Possess a Strong Th1-Polarization Capacity

Subsequently, we wanted to assess the functional capacity of zDCs to induce T cell polarization. For this purpose, we cultured immature DCs, αDCs, or zDCs with isolated naïve CD4+CD45RA+ T cells, as described in the Materials and Methods section. Co-cultures were performed in 48-well tissue culture plates. In all cases, RPMI supplemented with 10% human AB serum was used as the culture medium. After 7 days, intracellular cytokine secretion of IL-2, IL-10, and IFN-γ was determined in stimulated T cells, as described in the Materials and Methods section (Figure 4A). While we detected a low percentage of IFN-γ+ T cell populations from co-cultures with immature DCs, as expected, both αDCs and zDCs substantially increased the percentage of IFN-γ+ T cells. We observed a general trend for an increased ability of zDCs in comparison αDCs to induce IFN-γ+ T cell populations; however, although statistically significant (*p* = 0.0477), the change was not extensive (Figure 4B). For comparison purposes, we also analyzed the capacity of differentially stimulated T cells to secrete IFN-γ, by determining its levels in culture supernatants. After 7 days, T cells from the equivalent co-cultures as before were washed extensively, re-seeded, and stimulated with anti-CD2/CD3/CD28 macrobeads for 24 h. Afterwards, IFN-γ in the supernatants was determined, as shown in Figure 4C. In this case, we also observed increased IFN-γ values secreted by T cells stimulated with zDCs; however, the *p* value calculated from the four independent experiments did not demonstrate significance (*p* = 0.063).

### 3.5. zDCs Exceed αDCs at Induction of Cytotoxic Immune Responses

In order to investigate the capacity of zDCs to initiate responses of the final effector arm of anti-tumor immunity, we performed different CD8+ T cell-associated assays. Firstly, we examined the capacity of variously matured DCs to induce the proliferation of allogeneic whole CD8+ T cells. Dendritic cells were prepared as described in the Materials and Methods section. Immature DCs, αDCs, and zDCs were used as stimulators in co-cultures with CD8+ T cells at a 1:10 DC:T cell ratio. T cell proliferation was measured on day 5 using liquid scintillation counting (Figure 5A). Surprisingly, the capacity of zDCs in comparison to αDCs to induce CD8+ T cell proliferation was more than three-fold greater. Further investigation revealed that in equivalent co-cultures, the levels of IFN-γ in culture supernatants, where zDCs were used as stimulators, were extensively induced in comparison to immature DCs, and also greatly increased when compared to co-cultures where αDCs were used as stimulators (Figure 5B). Finally, we assessed how variously matured DCs can induce cytotoxic T cell activation in terms of intracellular granzyme B expression. Detection of granzyme B was analyzed after 5 days of co-culture between immature DCs, αDCs, and zDCs, as described in the Materials and Methods section (Figure 5C). The percentage of CD8+granzyme B+ cells was generally two times lower in cultures with immature DCs than with mature DCs. In line with the zDC capacity to induce superior CD8+ T cell proliferation and IFN-γ secretion, they also induced the greatest induction of granzyme B expression.

Finally, we examined the capacity of zDCs to induce Ag-specific CD8+ T cell effectors, directed against melanoma-associated Ags. Consistent with their increased capacity to produce IL-12p70 and activate CD8+ T cells (Figure 5), zDCs had the greatest capacity, in comparison to iDCs and αDCs, to induce an Ag-specific response (Figure 6) against Ags gp100 (154–162), gp100 (209–217), tyorsinase (369–377), and melan-A (26–35), as determined by counting the number of spot-forming colonies using the ELISPOT method, as described in the Materials and Methods.

## 4. Discussion

In the present manuscript, we present a novel DC maturation approach that represents a significant upgrade from the latest maturation protocols in terms of type-1 DC characteristics required for effective anti-tumor immunity. Despite their limited clinical efficacy in the past, the search for “superior” DC vaccines continues, and is backed by several recent advances in our understanding of what the future of anti-tumor therapy will be. Most likely, we will see more combination approaches (e.g., cell therapy in combination with immune checkpoint inhibitors), as well as a focus on the refinement of previously neglected factors, such as type of Ag used for DC loading, loading strategy, type of vaccine delivery, and also the potential need to apply various DC subsets in vaccination strategies [17,18,19,20]. Regardless of DC type, their activation toward full maturation status will be necessary for ensuring optimal immunogenicity and the prevention of tumor Ags. This is of particular importance for MoDCs, which lack some of the superior immune-activating properties of natural DCs, such as optimal production of IL-12 [21]. In practice, this is all the more relevant, since the majority of clinical studies rely on MoDCs for their clinical vaccine manufacturing process.

The majority of Ag-specific anti-tumor immune responses are carried out by type-1 cellular effectors, namely CD4+ Th1 cells and CD8+ CTLs. Both effector cell types are induced from naïve T cells after Ag recognition and activation by Ag-presenting cells. To this day, IL-12 produced by DCs is recognized as the most important contributor to the type-1 polarization process [22]. It causes the upregulation of the transcription factor T-bet in CD4+ and CD8+ T cells, thereby promoting their differentiation into Th1 and CTL cells, as well as their capacity to produce IFN-γ, which further strengthens their type-1 commitment and activation. For this reason, many of DC maturation protocols developed in the past were focused on how to utilize various DC-activation signals to strengthen their IL-12 secreting capacity. The latest major discovery in this context was made by the work of Kalinski and colleagues, who more than 10 years ago, have developed a unique DC maturation cocktail, enabling superior IL-12-producing capacity, consisting of TLR3 agonist poly I:C in combination with TNFα, IFNα, IFN-γ, and IL-1β [7]. We now show that, by upgrading this protocol by the inclusion of combined TLR signaling between TLR3 and TLR8 pathways, the IL-12-secreting capacity of DCs can be greatly enhanced. We refer to the mature DCs obtained in this manner as zDCs (maturation cocktail consisting of poly I:C, R848, TNFα, IFNα, IFN-γ, and IL-1β), and also compared their various functional capacities with αDCs. A crucial step in the zDC activation protocol is the pre-conditioning of DCs with poly I:C, prior to the addition of other components. A hypothesis for this step was based on a previous finding by Pohar et al., which demonstrated the importance of the UNC93B1 protein in HEK293 cells transduced with various TLRs, for the priming effect of poly I:C for subsequent stimulation by other nucleic acid agonists. Pohar et al. have demonstrated that pre-treatment with poly I:C stabilizes other nucleic acid-sensing TLRs (TLR7-TLR9) and strongly augments subsequent TLR9 responses [13]. In our preliminary experiments, we also confirmed this effect in the context of DCs, where the increased production of IL-12p70 could be observed in the case of the poly I:C pre-treatment, followed by the stimulation of TLR8, but not the other way around (data not shown). In this manner, IL-12p70 production directly after activation can be more than two-fold greater than with αDCs, or approximately three-fold greater than with DCs stimulated using LPS and IFN-γ (Figure 1A). The importance of poly I:C pre-treatment can be seen from the testing of individual components. While poly I:C can be regarded as a strong inducer of IL-12p70 production, we observed little or no IL-12p70 in the supernatants of DCs stimulated with R848 alone (Figure 1A). When poly I:C and R848 were used simultaneously, an increased capacity in IL-12p70 production was suddenly observed, particularly in samples that underwent poly I:C pre-treatment. The addition of type-1 cytokines to these samples strongly increased IL-12p70 production; this production was greatest in cultures with zDCs.

However, the secretion of IL-12 directly into culture supernatants does not necessarily reflect the capacity of DCs to produce IL-12 when it matters most, during the formation of immunological synapses, when DCs are re-stimulated by the CD40-ligand expressed on responding T cells [23]. We performed parallel experiments in which, after activation, the DCs were washed extensively and re-stimulated via the CD40/CD40-ligand pathway (Figure 1B). As expected, the quantitative levels of IL-12 fell after re-stimulation. The αDCs produced substantial quantities of IL-12 (ca. 1 ng/mL), which were comparable to DCs matured with LPS/IFN-γ and in similar a concentration range as shown previously by Wieckowski et al. [9]. Interestingly, zDCs were exceptional in this regard and greatly outperformed in all the tested protocols, producing more than eight-fold greater IL-12p70 levels than αDCs (Figure 1B).

To observe the quality of zDC maturation from a more uniform standpoint, we also performed a phenotype analysis, determining the expression of relevant co-stimulatory molecules (Figure 2). In comparison to known protocols, zDCs displayed a fully mature phenotype, expressing high levels of CD40, CD80, and CD86. However, we did not observe notable differences in expression compared to αDCs. The zDC also displayed positive expression of the chemokine receptor CCR7, and strongly upregulated MHC class II expression, the latter being greater in comparison to αDCs.

One of key DC vaccine characteristics is its responsiveness to CCR7-guided migration, and appropriate maturation approaches can be of crucial importance in this matter [24]. We observed the migratory capacity of variously matured DCs, using a transwell migration assay, toward the increased concentration gradient of chemokine CCL21. Although the expression of CCR7 on zDCs was not higher in comparison to αDCs or the other variants of maturation used, zDCs displayed significantly greater migratory efficiency, both in comparison to αDCs and DCs, stimulated with LPS/IFN-γ (Figure 3A). We do not believe that the observed disparity between CCR7 expression and CCR7-guided migration efficiency is unusual. Namely, the regulation of DC migration by CCR7 is not straightforward. It includes multiple signaling pathways, such as MAPK and RhoA, that regulate both chemotaxis and migratory speed [25]. Therefore, it encompasses a complex biology, most likely affected by crucial DC lifecycle events, such as maturation, and fine-tuned by the specific activation signals that trigger it. Efficient CCR7-directed migration could play an important role in optimizing anti-tumor T cell responses in vivo due to the increased lymph node homing of DC vaccine. The T cell stimulatory capacity of zDCs was tested in a 5-day MLR with whole CD4+ T cells. In this aspect, the zDC allo-stimulatory capacity was significantly higher in comparison to αDCs (Figure 3B). We further tested the capacity of zDCs to stimulate type-1 polarized responses of CD4+ T cells by performing co-cultures using naïve CD4+CD45RA+ T cells. After 7 days, the T cells were re-stimulated and stained intracellularly for the production of IL-2, IL-10, and IFN-γ (Figure 4A). With the exception of iDCs, both αDCs and zDCs induced T cell populations with typical type-1 polarization profiles, consisting of an increased percentage of IL-2- and IFN-γ-producing T cells. Throughout our experiments, we observed a slightly increased percentage of IFN-γ+ T cells in co-cultures with zDCs, which showed significance compared to the percentage of IFN-γ+ T cells induced by αDCs (Figure 4B). To further confirm the zDC T cell polarization characteristics, we performed parallel experiments in which, after the co-cultures, the T cells were washed and re-stimulated for 24 h, and secreted IFN-γ was measured in cell culture supernatants (Figure 4C). The results were similar to the intracellular IFN-γ-staining experiments, with increased IFN-γ production by zDC-stimulated T cells; however, we could not show significance when compared to αDCs (*p* = 0.063). Nevertheless, considering all results, we speculate that zDCs are slightly more efficient than αDCs at inducing Th1 effector cells.

For the main functional assessment of zDCs, we aimed to evaluate their capacity in relation to CD8+ T cell activation. The proliferation of whole CD8+ T cells was measured during a 5-day MLR, where the T cells were stimulated with iDCs, αDCs, or zDCs. Interestingly, the allo-stimulatory efficiency of zDCs exceeded that of αDCs by a great margin (Figure 5A). The increased level of activation of induced CTL effectors was reflected by the significantly greater IFN-γ levels in the co-culture supernatants (Figure 5B). In parallel experiments, the activation of CTLs was assessed by their intracellular expression of granzyme B (Figure 5C,D). We observed a substantial and significant increase in the percentage of granzyme B-positive CD8+ T cells in co-cultures in which zDCs were used as stimulators, in comparison to αDCs. In addition, this result rules out the possibility that CTL activation, measured by increased IFN-γ production (Figure 5B), is merely due to increased CD8+ T cell proliferation and the greater number of CTLs present in the culture. Particularly, the level of induction of Ag-specific CTLs directed at various tumor-associated peptides is an important part of the in vitro assessment of the DC vaccine potential, as well as the current standard of cancer patient immune monitoring after vaccine administration [26,27,28,29]. We assessed the capacity of zDCs to induce Ag-specific CTL responses aimed at melanoma-associated peptides derived from proteins gp100, tyrosinase, and melan-A. Consistent with our results describing the effect of zDCs on CD8+ T cell activation, proliferation, and IFN-γ production upon short-term stimulation, the long-term co-cultures (14 days), required to activate and expand the lower frequencies of Ag-specific T cell clones, demonstrated a superior capacity of zDCs, compared to αDCs, to expand the functional populations of melanoma-specific CTLs (Figure 6).

## 5. Conclusions

In conclusion, we present novel evidence that sets an additional benchmark in the pursuit of fully maximizing DC potential for use as a valuable tool for anti-cancer vaccine development. Although our combined results of zDC potency to elicit superior cytotoxic T cell responsiveness is not a direct guarantee, we expect to see the same effects in future in vivo studies; thus, we consider it as a basis for maturation protocols of the next generation. A great number of factors will determine the future success of DC-based cellular therapies, such as the further development of combination therapies, Ag selection, loading strategy, choosing the optimal DC type, and many others. However, optimal DC maturation will most likely play a role, and represents an important piece of the puzzle for ensuring maximum anti-tumor cytotoxic T cell responses and minimal immune cell exhaustion after administration.

## Figures and Tables

**Figure 1 cells-11-00835-f001:**
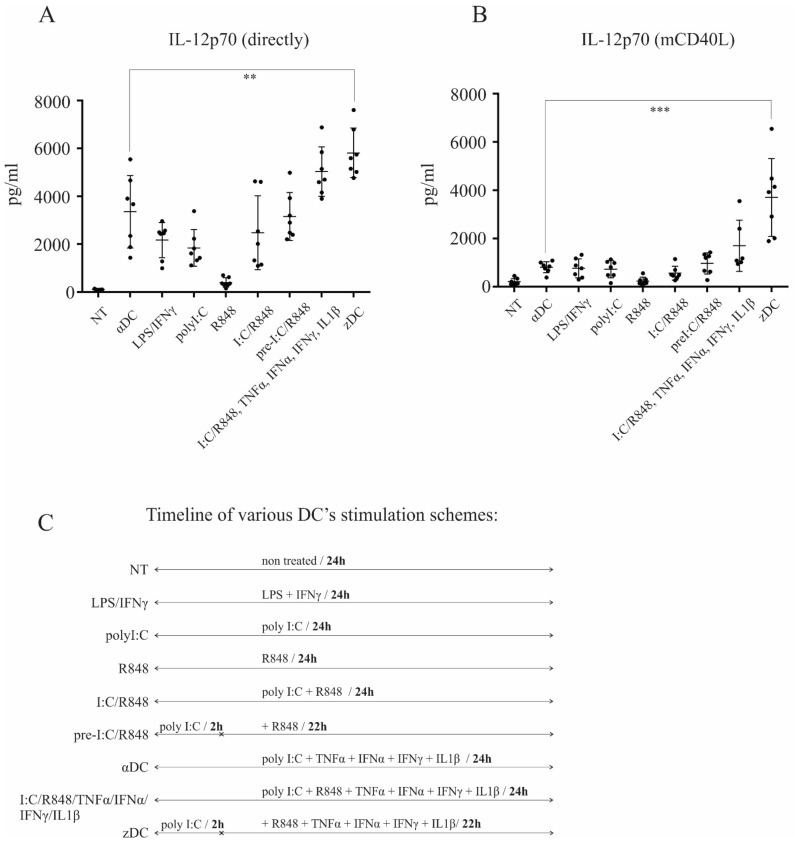
**zDCs possess extensive capacity for IL-12p70 production upon CD40-ligand stimulation.** Dendritic cells were generated from peripheral blood monocytes, as described in the Materials and Methods. (**A**) The production of IL-12p70 was measured directly from culture supernatants, after 24 h stimulation of DCs using various maturation protocols, as depicted in the figure. (**B**) IL-12p70 was measured from co-cultures after DCs were activated, extensively washed, and re-stimulated via CD40-CD40L pathway. Data represent mean ± SD of seven independent experiments. The black dots represent individual sample values. Statistical significance for comparison of zDC vs. αDC samples was performed using Student’s unpaired *t* test. A *p* value of less than 0.05 was considered statistically significant. (** *p* < 0.01; *** *p* < 0.001). (**C**) A timeline of various DC’s maturation schemes.

**Figure 2 cells-11-00835-f002:**
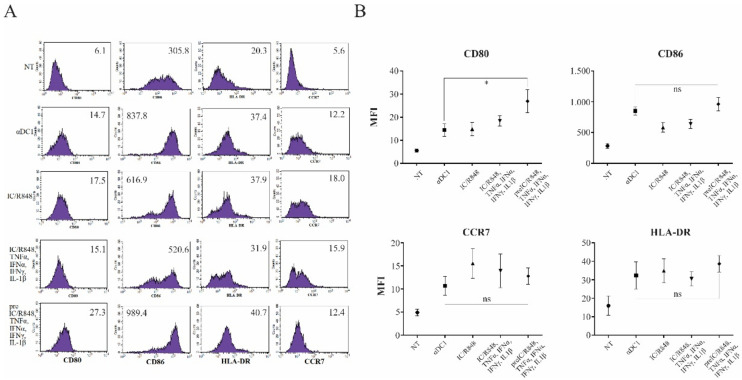
**zDCs display optimally matured phenotype.** (**A**,**B**) Dendritic cells were differentiated from monocytes isolated from freshly prepared human buffy coats, as described in the Materials and Methods. They were then either left untreated (NT, immature DCs) or stimulated for 24 h using different maturation protocols, focusing on various combinations of TLR3 and TLR8 agonists and inflammatory cytokines. After maturation, DCs were analyzed for the surface expression of CD80, CD86, CCR7, and HLA class II molecules. Numbers in histograms represent the mean fluorescence intensity values. Shown is one representative out of four independent experiments performed. Data were analyzed on a FACSCalibur system using CellQuest software version 3.3 (BD biosciences). Statistical significance between individual pairs was analyzed using Student’s unpaired *t*-test (ns—non significant; *—*p* < 0.05).

**Figure 3 cells-11-00835-f003:**
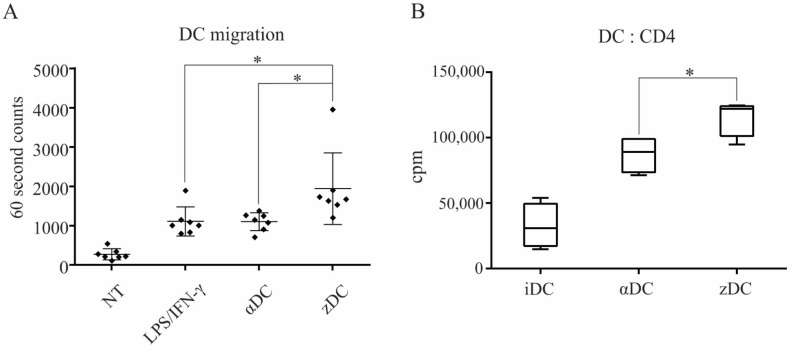
**zDCs are potent stimulators of T cell proliferation and have an efficient CCR7-guided migratory capacity.** (**A**) The migratory capacity of DCs, matured with different protocols to obtain either αDCs or zDCs, or activated using a combination of LPS and IFN-γ, was assessed in a transwell migration assay. Non-treated, immature DCs served as negative control. Lower chambers of the transwell contained 200 ng/mL of recombinant CCL21 protein and the assay was performed for 3 h at 37°C, 5% CO_2_. The number of DCs that migrated to the lower chamber was analyzed by flow cytometry using 60 s counts. The results shown are mean ± SD of 7 independent experiments. The black squares represent individual sample values. (**B**) The allo-stimulatory capacity of whole CD4+ T cells by variously treated DCs was assessed. DC:T cell co-cultures were set-up in 96 wells in a 1:10 ratio (2 × 10^4^ DCs, 2 × 10^5^ T cells), in 200 µL of RPMI + 10%AB serum. On day 5, cell proliferation was measured by liquid scintillation counting. The results shown are mean ± SD of counts per minute (cpm) of four independent experiments. Statistical significance between individual pair of samples was calculated using Student’s unpaired *t* test. A *p* value of less than 0.05 was considered statistically significant (*—*p* < 0.05).

**Figure 4 cells-11-00835-f004:**
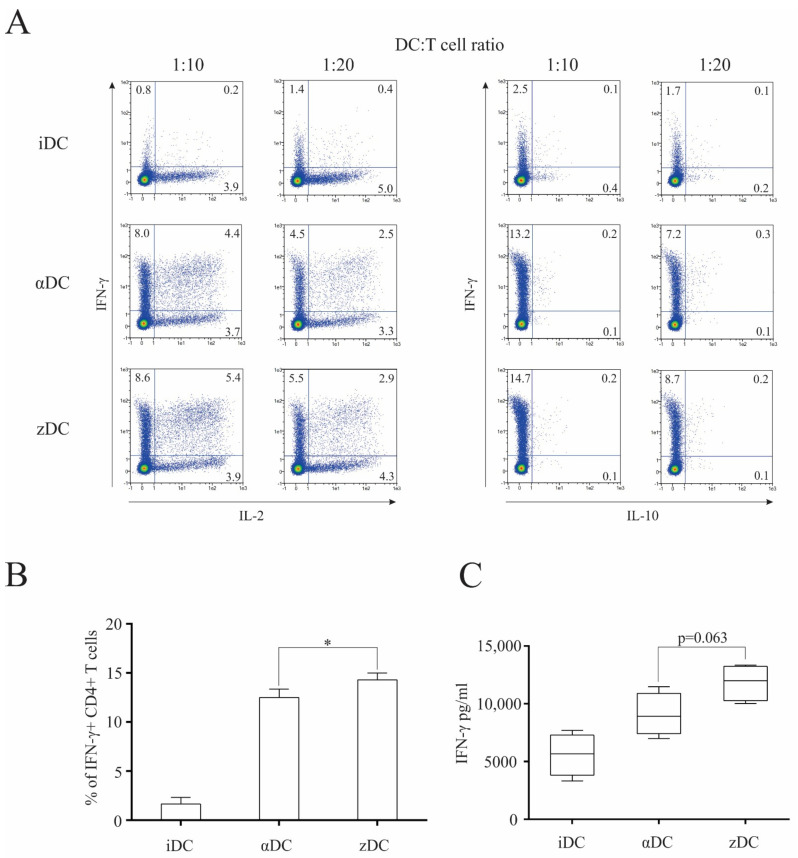
**zDCs are efficient at inducing Th1 polarization of naïve CD4+ T cells.** Dendritic cells were prepared as described in the Materials and Methods. (**A**) Their T cell polarization capacity was determined by performing co-cultures of variously treated DCs with allogeneic, naïve CD4+CD45RA+ T cells (48-well plates, 2 × 10^5^ DCs with 2 × 10^6^ naïve T cells) in RPMI medium supplemented with 10% human AB serum. After 7 days, the T cells were collected, re-stimulated, and stained intracellularly against IL-2, IL-10, and IFN-γ. Numbers in quadrants represent the percentage of positive cells. Shown is one representative out of four independent experiments performed. (**B**) Statistical analysis of four independent experiments, comparing percentage of IFN-γ-producing T cell populations (DC:T cell ratio 1:10). (**C**) In parallel experiments, similar identical co-cultures were performed as above. After 7 days, the T cells were collected, washed, and re-stimulated using anti-CD2/CD3/CD28 macrobeads. After 24 h cell culture supernatants were analyzed for the presence of IFN-γ. Data are shown as mean ± SD of four independent experiments. A *p* value of less than 0.05 was considered statistically significant (*—*p* < 0.05).

**Figure 5 cells-11-00835-f005:**
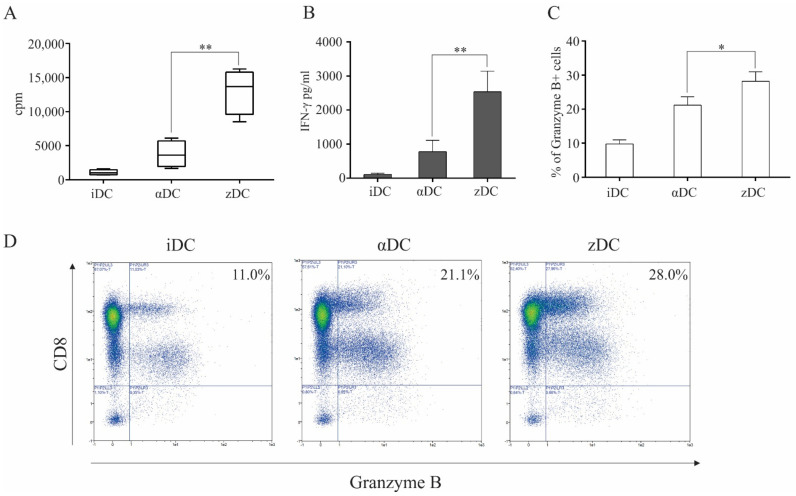
**zDCs have an exceptional capacity to induce cytotoxic T cell responses.** (**A**) iDCs, αDCs, or zDCs were used in co-cultures with whole, allogeneic CD8+ T cells. They were placed in 96-well tissue plates, with a 1:10 DC:T cell ratio. On day 4, the wells were pulsed with 1 μCi/well of 3H-thymidine. On day 5, cell proliferation was measured using liquid scintillation counting. Results show mean ± SD of four independent experiments (cpm, counts per minute). (**B**) In parallel co-cultures with allogeneic CD8+ T cells and variously treated DCs, after 5 days, levels of IFN-γ were analyzed in culture supernatants. Bar graphs represent mean ± SD of four independent experiments. Statistical comparison between individual samples was performed using Student’s *t* test. A *p* value of less than 0.05 was considered statistically significant (**—*p* < 0.01). (**C**,**D**) Cytotoxic T cell activation was measured by detection of intracellular granzyme B expression. iDCs, αDCs, or zDCs were used as stimulators of allogeneic CD8+ T cells. The co-cultures were performed in 48-well tissue plates in complete RPMI medium (10% human AB serum). After 5 days, the T cells were collected, washed, and stained for CD8 and granzyme B expression (intracellularly). Statistical comparison of individual pairs of three independent experiments was performed using Student’s *t* test (* *p* < 0.05). The numbers in upper right quadrants represent the percentage of granzyme B-positive CD8+ T cells.

**Figure 6 cells-11-00835-f006:**
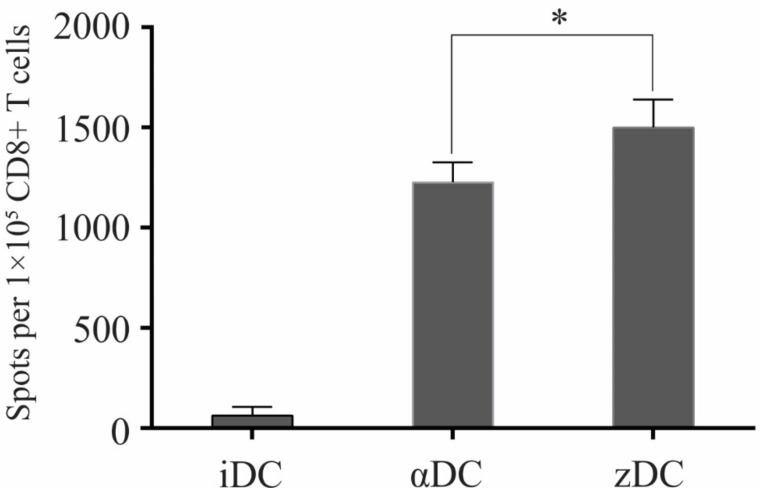
**zDCs have a superior capacity to induce melanoma-specific CD8+ T cell responses.** We sensitized autologous CD8+ T cells using two rounds of stimulation with iDCs, αDCs, or zDCs. The DCs were pulsed in advance with melanoma-associated peptides gp100 (154–162), gp100 (209–217), tyrosinase (369–377), and melan-A (26–35). After 14 days in DC:T cell co-culture, the T cells were harvested and re-stimulated with T2 cells as stimulators (previously pulsed with peptides) on pre-coated, IFN-γ ELISPOT strips for 24 h. The number of spot-forming colonies was determined and analyzed using an ELISPOT reader. The data are shown as mean ± SD of duplicates of three independent experiments, involving CD8+ T cells and DCs from different donors. Statistical significance between individual pairs was calculated using Student’s unpaired *t* test (* *p* < 0.05).

## Data Availability

Not applicable.

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
