# Peer review of "Combined TLR-3/TLR-8 Signaling in the Presence of α-Type-1 Cytokines Represents a Novel and Potent Dendritic Cell Type-1, Anti-Cancer Maturation Protocol"

_cells, 2022, doi:10.3390/cells11050835_

Round 1

Reviewer 1 Report

Fevžer et al propose a new protocol for DC maturation which can be very interesting in order to improve DC-based vaccines. The results presented by the authors demonstrate a synergistic effect of TLR-3 and TLR-8 stimulation in DC maturation in presence of α-type 1 maturation cocktail.

Minor points:

  1. The manuscript could be improved by including more data about the T cell cytolytic capacity induced by different mature DCs (αDCs and zDCs).
  2. How long DCs were matured for? It is not clear in the manuscript if they were 24 or 48h.
  3. How many independent experiments have been performed to prove a greater induction of granzyme B expression by zDCs. It would be better to also provide a statistical analysis.

Reviewer 2 Report

In this article, authors proposed a new protocol for dendritic cells stimulation involving pre-incubation with TLR-3 agonist (poly I:C), α-type maturation cocktail and TLR-8 stimulation by R848. These DCs exhibited higher CD80 expression, IL-12p70 production,  greater migratory capacity toward chemokine ligands (CCL)19 and CCL21 and 27, greater allo-stimulatory capacity, induce more CD8+ T cell proliferation and cytotoxic T cells, which produced more Granzyme B. The article is of good quality, and results can have significant impact on the development of anti-tumor DC-based vaccines as authors proposed; however, the anti-tumor capacity of this strategy was not evaluated. The manuscript is well-written, however the discussion of the data should be properly discussed and compared with previous research. Materials and methods are precisely described, thus it would ensure reproducibility of experiments. Here major comments:

Comment #1. The title “Next-generation, anti-tumor dendritic cell type 1 maturation protocol requires TLR-3/TLR-8 synergy in the presence of alpha-type-1 cytokines” does not reflect the proper content of the manuscript. Firstly, and most important, authors did not evaluate the anti-tumor capacity of stimulated DCs. In addition, authors should clarify in the manuscript the reason for proposing this DCs stimulation strategy as a “next-generation” protocol. As reference, the cited the review of Garg et al, where they introduced the term “next-generation DCs vaccines” to refer to those vaccines that include DC subsets naturally occurring DCs, cancer neoantigens, and/or ICD to improve CD8+ T cells–driven anticancer immunity. This definition is not consistent with the experimental procedure evaluated here, which seems more an alternative to “second-generation” DCs vaccines that improved mo-DCs stimulation via maturation cocktails.

Comment #2. In materials and methods, authors must indicate the information regarding the statistical analysis they perform.

Comment #3. When introduced the experimental design, in order to clarify the variables, authors should include a “time-line” scheme to show the different stimulation protocols that they compared in the experiments.

Comment #4. It is important to include an explanation of the molecular mechanism of polyI:C and R848 as TLR3 and TLR8 ligand, respectively.

Commnet #5. In figure 4A and 5C, authors showed dot plots showing a representative result. Please, include the graph (bar or dots) and the statistical analysis associated to these data.

Comment #6. Along the manuscript, authors propose that combination between TLR3 and TLR8 stimulation plus alpha-cytokines promotes synergistic stimulation of dendritic cells (e.g., lines 225-227, line 394). This reviewer has a few concerns whit those statements, because authors did not properly evaluate synergism between them. I recommend, considering Chou-Talalay methods, to evaluate the effect of the combinations, in order to classify them as additive, synergic or antagonist (DOI: 10.1158/0008-5472.CAN-09-1947).

Comment #7. The evidence here seems far too speculative to propose that zDCs possess a strong Th1-polarization capacity. Data included in Figure 4 do not show statistical difference.

Comment #8. Author says: “A crucial step in zDC activation protocol is the pre-conditioning of DCs with poly I:C, before the addition of other components” (387-388). Authors should explain the rationale behind this experimental procedure and demonstrate than pre-conditioning with TLR3 ligand is better that pre-conditioning with TLR8 ligand.

Comment #9. In discussion session, the results are not discussed from multiple angles and placed into context, it seems a repetitive list of the observed results; I consider authors should improve this. For example, compare with other dendritic cells` stimulation protocols previously reported or hypothesize the molecular basis of the response that they report.

Reviewer 3 Report

Dear Authors,

Thank you for the chance to review your paper describing the role of synergism of TLR3/8 in generation of more potent anticancer DC maturation.

In general the whole study is smooth, reasonable, and fairly executed, albeit, it lacks novelty and more self-criticism.

Firstly,, I would like to point out that there are many, many similar (not the same although) studies executed/published recently. There are many different pathways/stimulation pattern studied. Definitely, if I had to review this paper in 2015 I would do this for high impacted journal. These days, the novelty drop and your results are not in the top as presented by others. (Please refer to works of Matsumoto et al 2015 and 2017; M Damo et al 2015, Y Wang et al 2020). And much more. So, the engineering of DC using TLRs is a common and highly studied phenomena. TLR3/8 signalling in DC as well. It would be good if you can prove more evidence of your findings using chemical agents modyfing studied TLRs. You based all your findings on very limited data and very limited experimental approach. However, we must pursue science forward.

My question is - what is the basic characteristics of donors of blood? How many different donors were involved? What age? Sex? CigSmoking status? Antibiotics in the past 2 weeks? It counts a lot - from my experience the lack of table summary with these information is a sign of not fully understood the nature of WBC in general and for in vitro/ex vivo studies.

What are the real chances that your method, not the one from 123 others will get the green light from pharma industry? How your study is superior to other? What is the reason for choosing synergy proposed by you, not the other?

And the another side of the coin - what are the limitations of your study? What are the pitfalls and what are the solutions when the pitfall occurs? You know, even best executed study might be biased. The job of me, your reviewer, is to make sure that your insight is fair and unbiased.

I do like to see FSC vs SSC charts as supplementary file for all (or at least 3) flow experiments for your monocytes purity assessment.

I wonder, how did you choose the 3/8 synergy? There is bunch of other combinations, why just this - did you try others? Did you do in silico preliminary engineering?

What was the repeatability for your experiments? 

Last thing, I doubt whether your percentage calculations for Fig 5C are correct for upper-right square. The figures, in general, look very fuzzy - I mean pretty low quality - needs improvement. But figure 5C looks like the values for U-R square are not correct. I almost not seeing scaling to green gating in the square. It might be because of something I am not aware, but 15 years working on flow give me some confidence in my assessment. 

Please feel free to discuss my review in an even sharp way and I hope to see the revised version soon,

All the best in the upcoming year,

Round 2

Reviewer 2 Report

The authors' response and revisions have satisfactorily addressed my comments #1, 5, 6, 7, 8 and 9 on the earlier version of the manuscript. They have included new data to support the conclusion, re-formatted the manuscript (including a new title) and improved the discussion, according to my previous comments, therefore I recommend publication. Nevertheless, I suggest the following minor changes:

Comment #2. Authors indicate that “Statistical significance between individual pairs was calculated using Student’s unpaired t-test” (lines 222-223). However, t-test is statistical hypothesis test used to compare the means of two population groups. In most cases, experiments have more than two population groups and it´s more appropriate to perform an ANOVA analysis.

Comment #3. Even when the explanation of authors is clear and consistent, the information in the manuscript is presented in a bit confused way and could be misunderstanding. I still suggest including a time-line showing the different stimulation´ schemes.

Comment #4. Authors included an excellent explanation about the rationale behind the combination proposed. However, this comment was addressed to show evidence that demonstrate that polyI:C is a TLR3 ligand (e.g. doi.org/10.1002/eji.201242902) and R848 a TLR8 ligand (e.g. doi.org/10.1016/j.vaccine.2005.06.024), and to explicit mention it in the text.

Author Response

Please see the attachment,

Reviewer 3 Report

Dear authors,

You made elegant work within the whole review process and I also recognize your expertise in this field. I am happy with the scientific discussion during the review as well as with your scientific approach to the publication process. The draft gained quality, as well as my major doubts, have disappeared. 

Right now, I am expecting you to fix only a few minors:

  • please add information in the statistical approach section whether you evaluated the normal/skewed distribution - if yes - how, if no - why?
  • line 324 - please specify a range of the trend - was it 0.05>p>0.1 or rather p>0.2? It is important to highlight it, even if not significant
  • the numbers in the reference section should be typical in a font size
  • line 165 please provide the country/city of origin of this manufacturer. The same for lines 135, 136 and so on.

Best.
